# Does Adherence to the Mediterranean Diet Have a Protective Effect against Asthma and Allergies in Children? A Systematic Review

**DOI:** 10.3390/nu14081618

**Published:** 2022-04-13

**Authors:** Despoina Koumpagioti, Barbara Boutopoulou, Dafni Moriki, Kostas N. Priftis, Konstantinos Douros

**Affiliations:** 1Department of Nursing, National and Kapodistrian University of Athens, 11527 Athens, Greece; despina.koumpagioti@gmail.com (D.K.); bmpoutopoulou@gmail.com (B.B.); 2Third Department of Pediatrics, Attikon University Hospital, School of Medicine, National and Kapodistrian University of Athens, 12462 Athens, Greece; dafnimoriki@yahoo.gr (D.M.); costasdouros@gmail.com (K.D.)

**Keywords:** Mediterranean diet, asthma, allergies, child

## Abstract

Dietary pattern may potentially impact on the pathogenesis of asthma and allergies. The Mediterranean Diet (MD) has significant health benefits due to its antioxidant and anti-inflammatory properties. The aim of this systematic review was to investigate the effectiveness of adherence to the MD against asthma and allergies in childhood. Hence, a systematic literature search was conducted on PubMed, ESBCO (Cinahl), Scopus, and the Cochrane Library databases up to 26 January 2022. The total number of articles obtained, after the initial search on the databases was conducted, was 301. Twelve studies were included, after the removal of duplicates and screening for eligibility. Our findings indicated a protective role of the MD against childhood asthma, but they also imply that the MD probably does not affect the development of allergies. Nevertheless, the heterogeneity and limitations of the studies highlight the need for randomized controlled trials that will focus on the pediatric population and hopefully provide more robust evidence.

## 1. Introduction

Asthma and allergies are increasingly prevalent worldwide among the pediatric population. Their etiology is multifactorial and relies on complex interactions between genetic and environmental factors, leading to a phenotypical manifestation of the disease [1,2]. Dietary intake pattern is considered to be a plausible environmental explanation and seems to have a fundamental role in the development of the microbiota which may influence the immune, inflammatory and allergic mechanisms [3,4]. 

Modern diets including the Western diet are based on a high intake of processed foods, sugar, saturated fats, and low fruit and vegetable consumption [5]. High adherence to Western dietary patterns has been linked with an increased risk of frequent respiratory symptoms in three- and four-years old children [6]. In contrast, the Mediterranean Diet (MD) is composed of a high intake of fruits, vegetables, cereals and olive oil; a moderate intake of white meat, fish and dairy products; and a low intake of sugar and red meats [5]. MD has been consistently associated with enhancing immune function and having antioxidants and anti-inflammatory activities due to its abundance in micro/macronutrients such as vitamins (A, C, D), minerals (iron, zinc, selenium, folate/folic acid) and fatty acids (monounsaturated and polyunsaturated omega 3 fatty acids) [7].

Several studies during the past few years have established an association between the role of MD and revealed its protective effect on childhood asthma and allergies [8,9,10]. The International Study of Asthma and Allergies in Childhood (ISAAC) showed that a combined index of potentially modifiable lifestyle factors, including adherence to MD, no parental smoking, healthy body mass, physical activity and non-sedentary behavior was inversely related to “current wheeze”, “asthma ever” and current symptoms of rhinoconjunctivitis and eczema, in children six- to seven-years old [11]. However, other studies, carried out in children and adolescents, supported no MD association with any asthma or allergy outcome [12,13,14,15]. Some explanations of these controversial results might include the difference between age groups in the pediatric population; sample size; study design; the variable diet; the variable result (wheezing, asthma, other allergic disorders, sensitization) and dietetic hypothesis [1,16]. Garcia-Marcos et al. conducted a systematic review and meta-analysis on the influence of the MD on asthma in children a few years ago [16]. The aim of the current systematic review was to investigate the effectiveness of adherence to MD on asthma and allergies in childhood by updating the previous review with the results of recent studies.

## 2. Materials and Methods

This systematic review was accomplished according to the PRISMA statement for reporting systematic reviews [17].

### 2.1. Literature Search and Study Selection

A systematic literature search was conducted on PubMed, ESBCO (Cinahl), Scopus and the Cochrane Library databases up to 26 January 2022, using the MeSH (Medical Subject Headings) terms “Mediterranean Diet”, “asthma”, “allergies”, and “child”. A structured search and study selection were performed by two independent reviewers (D.K. and B.B.), with all differences being resolved by consensus.

### 2.2. Inclusion and Exclusion Criteria

#### 2.2.1. Inclusion Criteria

Inclusion criteria were: (i) Studies that were published in the English language; (ii) Studies that were published from 1 January 2012 until January 2022; (iii) Studies with target population children (up to 1 year-old) and/or adolescents; (iv) Studies that assessed and included distinct outcomes of the effect of MD on asthma and allergies.

#### 2.2.2. Exclusion Criteria

Exclusion criteria were: (i) Studies that evaluated other dietary patterns; (ii) Studies that assessed maternal nutrition exclusively; (iii) Studies with target population adults, pregnant women or infants; (iv) Studies that contained inadequate data according to dietary pattern or effect of MD on asthma and allergies.

### 2.3. Data Extraction

Two authors (D.K. and B.B.) extracted data from the selected studies, by applying inclusion/exclusion criteria, using: the author’s name, publication year, country, study design, sample size and age, MD score and the effect of MD on asthma and/or allergies.

## 3. Results

### 3.1. Selection of Studies

The databases’ search retrieved 301 articles. Of these, 126 were excluded because of duplicates and reviews. The remaining 175 original ones were screened for relevance. The full-text of 44 articles were assessed for eligibility and, finally, 12 studies met the inclusion criteria and were selected for the systematic review. The detailed selection process is depicted in Figure 1.

### 3.2. Characteristics of Studies

A total of 12 studies that investigated the effect of adherence to MD on asthma and allergies in children were included in the systematic review. Three studies were conducted in Greece [9,18,19], three in Turkey [13,14,20], two in Spain [21,22], one in France [8], one in Peru [12], one in Brazil [23], and one in Lebanon [10].

Seven studies were cross sectional studies [9,10,13,14,18,20,23], one randomized controlled trial [19], one case-control [12], one cohort [8], one prospective before-after comparison study and one prospective longitudinal study [22].

The total sample size of children and adolescents of the 12 studies, aged 1 to 19 years old, was 34,972. The sample sizes ranged from 64 to 9991 participants.

Adherence to MD was measured by diet quality indices. Four studies [9,18,19,21] used the KIDMED (Mediterranean Diet Quality Index for children and adolescents) index developed by Serra-Majem et al. [24,25], five studies [12,13,14,20,22] used the Mediterranean Diet Score developed by Garcia-Marcos et al. [26] and modified by Psaltopoulou et al. [27], whilst one study [8] used both KIDMED [24] and the Mediterranean Diet Score (MDS), developed by Trichopoulou [28]. Furthermore, one study [23] measured the adherence to MD qualitatively, with a frequent intake of at least 5 foods in each group to be classified as “yes” [29] and another study [10] used Food Frequency Questionnaire (FFQ) [30,31] for assessing participants’ dietary habits, and the group of food items including fish and olive oil was called MD.

The majority of the studies reported the effect of adherence to MD on asthma [9,10,14,18,19,21,23], three studies reported the effect on both asthma and allergies [8,12,22], and two studies on allergies [13,20]. The main studies’ characteristics are presented in Table 1.

### 3.3. Effect of Adherence to MD on Asthma in Children

Ten out of twelve studies evaluated the effect of adherence to MD on asthma in children [8,9,10,12,14,18,19,21,22,23]. Summarizing the outcomes, most of the studies identified a protective role of MD against childhood asthma.

Amazouz et al. found that children with a high adherence to MD, as measured by the KIDMED score, had significantly higher FEV_1_ (Forced Expiratory Volume in 1 s) (adjusted beta coefficient (aβ) = 52.3 mL, 95% CI: 5.5–99.1) and FVC (Forced Vital Capacity) (aβ = 67.4 mL, 95% CI: 11.6–123.3) and a lower risk of current asthma (adjusted odds ratio (aOR = 0.19, 95% CI: 0.04–0.85). When adherence was estimated with the MDS, the results were similar with children in the high adherence group having higher FEV_1_ (aβ = 49.9 mL, 95% CI: 15.1–84.6) and FVC (aβ = 69.7 mL, 95% CI: 27.9–111.6) and a lower risk of current asthma (aOR = 0.28, 95% CI: 0.12–0.64) [8]. A randomized controlled trial revealed that MD, supplemented with two fatty fish per week, reduced airway inflammation in asthma (*p* = 0.04), as assessed by Fractional Exhaled Nitric Oxide (ppb) [19]. Douros et al. showed that adherence to MD was related to better regulation of the main inflammatory mediators of asthma IL-4 (*p* = 0.007), IL-33 (*p* = 0.010) and IL-17 (*p* = 0.017) [18]. Malaeb et al. documented that occasional and daily MD consumption was significantly associated with lower odds of current asthma (*p* = 0.002 and *p* = 0.005, respectively) [10]. Additionally, three more studies showed the beneficial role of adherence to MD on asthma, reporting decreased odds of asthma (*p* = 0.02) [12], fewer asthma attacks, decreased use of ICS (inhaled corticosteroids) and SABA (Short–acting beta_2_ agonists) (*p* < 0.001) [21] and fewer asthma symptoms (*p* < 0.001) [9].

On the contrary, three studies found no significant associations between adherence to MD and asthma symptoms (*p* = 0.85) [14], asthma severity (*p* = 0.40) [23] or current wheezing in children four-years old (*p* = 0.44) [22].

### 3.4. Effect of Adherence to MD on Allergies in Children

Five studies assessed the effect of adherence to MD on allergies in children [8,12,13,20,22]. Amazouz et al. found that children with high adherence to MD, as measured by the KIDMED score, had a lower risk of any allergen sensitization (aOR = 0.56, 95% CI: 0.32–0.99), to food allergens (aOR = 0.38, 95% CI: 0.15–0.94) and to inhalant allergens (aOR = 0.60, 95% CI: 0.33–1.08). When adherence was estimated with the MDS the results were similar, with children in the high adherence group having a lower risk of sensitization to any allergens (aOR = 0.59, 95% CI: 0.39–0.90) or sensitization to inhalant allergens (aOR = 0.53, 95% CI: 0.34–0.82). However, no significant association was found between adherence (measured with either KIDMED or MDS) and current rhinitis or current eczema [8]. Tamay et al. noted no significant effect of MD on lifetime rhinitis (*p* = 0.78), physician-diagnosed allergic rhinitis (*p* = 0.63) or current rhinoconjunctivitis (*p* = 0.06) [20]. Moreover, three more studies showed that MD adherence did not remain a protective factor for allergic rhinitis (*p* = 0.18) [12], (*p* = 0.30) [13], (*p* = 0.096) [22] or atopic status (*p* = 0.49) [12].

## 4. Discussion

The present systematic review investigated the effect of adherence to MD on asthma and allergies in children. The vast majority of studies identified the protective role of adherence to MD against asthma, reporting higher spirometric indices and lower risk of current asthma [8], reduction of airway inflammation [19], better regulation of asthma-related interleukins [18] and limited asthma symptoms [9,10,12,21]. Concerning allergies, most studies supported no significant effect of MD on allergic rhinitis, current eczema, current rhinoconjunctivitis or atopic status [8,12,13,20,21].

Our findings, regarding childhood asthma, are in concordance with other systematic reviews [16,32,33,34,35] and meta-analyses [16,33,34,35], assessing the role of MD on asthma. Garcia-Marcos et al. supported that MD adherence was associated with a lower occurrence of “asthma ever”, “current wheeze” and “current severe wheeze” in children, with the results for “current wheeze” and “current severe wheeze” to be driven by Mediterranean regions [16]. Papamichael et al. found an inverse association between adherence to MD pattern and asthma in children, despite the absence of randomized controlled trials [32]. Zhang et al. suggested that there was an inverse association between high adherence to the Mediterranean diet during childhood and the risk of current wheeze [33]. Another two meta-analyses documented that MD was protective for persistent wheeze [34], childhood asthma and/or wheeze [35]. Furthermore, a review of Castro-Rodriguez et al. updated the evidence on the effect of adherence to MD on asthma, allergic rhinitis and atopic eczema in childhood, and showed that MD had a protective effect on asthma/wheezing symptoms, after adjustment for confounders. However, the impact of MD on lung function and bronchial hyperresponsiveness was doubtful [36].

Research findings have justified MD’s key role in asthma prevention. MD is based on a variety of fruits, vegetables and wholegrain cereals [2]. Nutrients including vitamins, fibers, minerals and fatty acids have been proven to possess anti-inflammatory properties and have a significant role in the protection of the respiratory tract [7,37]. MD can enhance the endothelial function through the improvement of pro-inflammatory markers, including the high-sensitivity C-reactive protein, IL-6, and adiponectin level, whilst its nutrients have been correlated with reduced bronchial hyperresponsiveness [38,39,40].

In contrast with the above findings, three studies revealed no significant associations between adherence to MD and asthma symptoms [14], asthma severity [23] or current wheezing in four years old children [22]. Castro-Rodriguez et al. suggested that the failure to show a protective role of MD in the development of wheezing relies on confounding factors such as parental rhinitis, birth weight below 2 kg, maternal tobacco use during pregnancy and mold stains, that blunt the MD effect [22]. Silviera et al. attributed the absence of significant association between MD and asthma severity to the study’s several limitations including the small sample size, the broad age range (3–12 years old) as a confounding factor in food consumption that might differ among age groups or the merging of mild persistent, moderate persistent and severe persistent asthma into one “persistent asthma” category and not examining them separately [23]. Moreover, Akcay et al. evaluated the prevalence of asthma and the risk factors affecting asthma symptoms in adolescents and found no association between MD and asthma prevalence. In their study, it is mentioned that participants had a high intake of fermented drinks made from millets and various seeds, mixed pickles, margarine, butter and meat, which was related to a high risk of asthma [14]. It is possible that the high consumption of processed food by adolescents counteracted the beneficial role of MD on asthma. In addition, it is widely known that the adoption of a dietary pattern during childhood, based on fast food, Western Diet and processed meat is related to the development of asthma in adult life [2].

In our systematic review, the majority of studies assessing the effect of MD on allergies in childhood found no significant association [8,12,13,20,22]. These findings coincide with Castro-Rodriguez’s et al. review that reported no significant effect of MD on preventing atopic eczema, rhinitis or atopy [36]. Tamay et al. assessed the prevalence of allergic rhinitis and its relationship with dietary habits and other risk factors among six- to seven-years old children; MD had a significant protective role only in the univariate analysis, whilst the multivariate analysis showed no association. This may be due to children with or without allergic rhinitis sharing similar dietary habits. Also, the combination of dietary habits with different environmental factors might have influenced the prevalence of allergic diseases [20]. Castro-Rodriguez et al. who investigated the effect of MD consumption, both during pregnancy and early in life, on current rhinitis and dermatitis at preschool age, noted no association, due to the presence of risk factors such as pets at birth; maternal rhinitis; higher maternal education level; increased weight and mold stains [22]. An exception to previous outcomes is the Amazouz et al. study which mentioned that high adherence to MD was associated with a lower risk of any allergy sensitization in school-aged children, although no significant association was found with current rhinitis or eczema [8]. In addition, the adoption of an antioxidant dietary pattern, such as MD, with increased b-carotene intake, was related to a reduced risk of allergic sensitization and lower IgE levels, in five and eight years old children [41]. In the current review, we did not examine the effects of MD on conditions related to atopy, such as celiac disease [42].

The present study had several limitations. There was heterogeneity among the included studies in the designs, sample sizes, tools assessing MD adherence, participants’ ages, variable outcomes (e.g., asthma, wheezing, atopy, sensitization, other allergic diseases) and adjusted confounders. Moreover, the lack of randomized controlled trials (the vast majority of studies were observational with cross-sectional design) renders the causal relationships uncertain, as has been noticed by other authors as well [16,32,33].

The systematic literature search of recently published data regarding the effect of MD adherence on asthma and allergies in childhood is one of this study’s strengths. Indeed, the literature has been lately enriched with more data through the emergence of recently published studies that have provided updated feedback in this field.

## 5. Conclusions

In conclusion, the present systematic review showed that adherence to MD seemed to have a protective role against childhood asthma, but no effect was found on allergic rhinitis, eczema or atopy. The remarkable heterogeneity and limitations of the studies highlight the need for randomized controlled trials that will focus on the pediatric population and hopefully provide more robust evidence.

## Figures and Tables

**Figure 1 nutrients-14-01618-f001:**
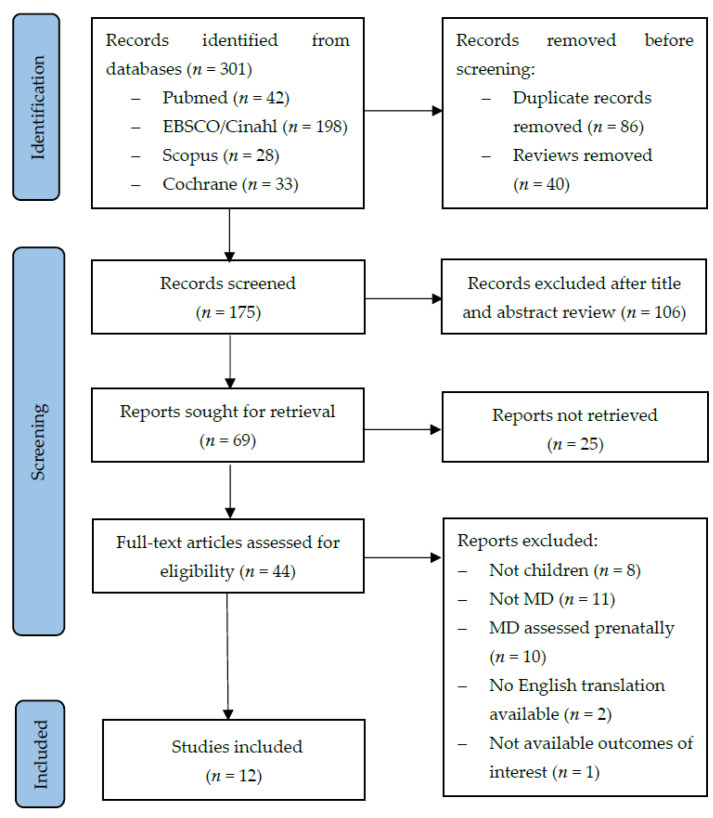
PRISMA diagram for study selection process.

**Table 1 nutrients-14-01618-t001:** Summary of studies’ characteristics.

Author	Country	Duration	Study Design	Sample Characteristics	MD Score	Asthma/Allergy Outcome	Results–Effect of MD	Covariates
Tamay et al. [13]	Turkey	Follow-up in 1 year	Cross-sectional	9991 13–14 years-old	MD Score (Range 0–22) [26,27]	Physician- diagnosed allergic rhinitis	No significant association with physician-diagnosed allergic rhinitis (*p* = 0.30)	Not adjusted
Akcay et al. [14]	Turkey	16 months	Cross-sectional	9991 13–14 years old	MD Score (Range 0–22) [26,27]	Physician-diagnosed asthma	No significant association with prevalence of asthma (*p* = 0.85)	Gender, family atopy history, residence, paracetamol use, parents’ educational level, domestic animals at home, siblings, television watching, tonsillectomy and adenoidectomy history
Tamay et al. [20]	Turkey	Follow-up in 1 year	Cross-sectional	9875 6–7 years old	MD Score (Range 0–22) [26,27]	Lifetime rhinitis, physician-diagnosed allergic rhinitis, current rhinoconjunctivitis	No significant association with lifetime rhinitis (*p* = 0.78), physician-diagnosed allergic rhinitis (*p* = 0.63), current rhinoconjunctivitis (*p* = 0.06) in the multivariate analysis	Gender, mother’s education, father’s education, exercise, television time
Antonogeorgos et al. [9]	Greece	48 months	Cross-sectional	1125 10–12 years old	KIDMED index (Range 0–12) [24,25]	Ever asthma symptoms	Significant negatively association with asthma symptoms (*p* < 0.001)	Age, gender, BMI, parental atopy
Rice et al. [12]	Peru	6 months	Case-control	383 9–19 years old 287 with asthma 96 controls	MD Score (Range 0–22) [26,27]	Current asthma, asthma control, FEV_1_, allergic rhinitis, atopic status	Significant association with decreased odds of asthma (*p* = 0.02). No association with asthma control (*p* = 0.3), FEV_1_ (*p* = 0.24), allergic rhinitis (*p* = 0.18), atopic status (*p* = 0.49)	Maternal education, age, sex, BMI
Silviera et al. [23]	Brazil	14 months	Cross-sectional	394 3–12 years old 268 with persistent asthma 126 controls with intermittent asthma	Qualitatively, with a frequent intake of at least five foods in each group to be classified as “yes”	Persistent asthma, intermittent asthma	No significant association between persistent and intermittent asthma (*p* = 0.40)	Not adjusted
Calatayud-Sáez et al. [21]	Spain	12 months	Prospective before-after comparison	104 1–5 years old	KIDMED index [24]	Wheezing, intensity of asthma attacks, BHR, cough, medication, infections, emergency room visits, hospital admissions	Significant association with the reduction of ICS and SABA need (*p* < 0.001). 32.2% free of asthma attacks, 35.3% only 1 attack/year, 24.9% 2 attacks/year vs. 4.73 attacks/year	Not adjusted
Castro-Rodriguez et al. [22]	Spain	Follow up at 1st and 4th year of life	Prospective longitudinal	1000 1.5 and 4 years old	MD Score (Range 14–36) [26,27]	Current wheezing, current rhinitis, current dermatitis at 4 years old	No protective factor for current wheezing (*p* = 0.44), current rhinitis (*p* = 0.096) or current dermatitis	Age, gender, maternal age, weight birth, breastfeeding, maternal studies, oral contraceptive use, siblings, paracetamol use during pregnancy, colds during first year of life, BMI, parental asthma/rhinitis/dermatitis, mold stains, physical activity, kindergarten, television-video
Malaeb et al. [10]	Lebanon	6 months	Cross-sectional	1000 Mean age 10.34 ± 3.96 years 807 healthy, 86 with probable asthma, 107 with current asthma	MD score based on FFQ [30,31], with the intake of fish and olive oil to be called MD	Current asthma	Adherence to MD (occasional and daily consumption) was associated with lower odds with current asthma (*p* = 0.002 and *p* = 0.005, respectively)	Sex, school type (public vs. private)
Douros et al. [18]	Greece		Cross-sectional	70 5–15 years old 44 with asthma 26 healthy	KIDMED index (Range 0–12) [24,25]	Asthma-related interleukins (IL-4, IL-33, IL-17)	Significant association with better regulation of inflammatory mediators IL-4 (*p* = 0.007), IL-33 (*p* = 0.010), IL-17 (*p* = 0.017)	Age, gender, BMI, SPT
Papamichael et al. [19]	Greece	6 months	Randomized controlled trial	64 5–12 years old 31 with asthma (MD plus two fatty fish/per week) 33 with asthma controls (usual diet)	KIDMED index (Range 0–12) [24]	Current asthma, spirometry, asthma control, FeNO	MD with two fatty fish/week was associated with a decrease in airway inflammation (FeNO) (*p* = 0.04). No significant association with asthma control and spirometry	Age, gender, physical activity, BMI
Amazouz et al. [8]	France		Cohort	975 8 years old	KIDMED index * (Range 0–14) [24,25] MD Score ** (Range 0–8) [28]	Current asthma, current rhinitis, current eczema, FEV_1_, FVC, FeNO	Significant association with higher FEV_1_ (*p* * = 0.06 and *p* ** = 0.04), FVC (*p* * = 0.04 and *p* ** = 0.01), lower risk of current asthma (*p* * = 0.05 and *p* ** = 0.01) and any allergen sensitization (*p* * = 0.04 and *p* ** = 0.02). No association with current rhinitis (*p* * = 0.39 and *p* ** = 0.79) or current eczema (*p* * = 0.11 and *p* ** = 0.82)	Maternal smoking during pregnancy, parental socioeconomic status at birth, parental history of allergies, older siblings at birth, sex, ethnicity, maternal origin, breastfeeding, exposure to tobacco smoke, consumption of organic food, physical activity, BMI

MD: Mediterranean Diet; BMI: Body Mass Index; FEV_1_: Forced Expiratory Volume in 1 s; ICS: Inhaled Corticosteroids; SABA: Short-activated beta_2_ agonists; FVC: Forced Vital Capacity; FeNO: Fraction Exhaled Nitric Oxide; SPT: Skin Prick Tests to common aeroallergens; BHR: Bronchial Hyperresponsiveness; FFQ: Food Frequency Questionnaire; * KIDMED index; ** MD Score.

## Data Availability

Not applicable.

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
