# Peer review of "Does Adherence to the Mediterranean Diet Have a Protective Effect against Asthma and Allergies in Children? A Systematic Review"

_nutrients, 2022, doi:10.3390/nu14081618_

Round 1

Reviewer 1 Report

Thank you for granting me the opportunity to review this piece of work. In this work, Koumpagioti et al. conducted a systematic review that investigated the association between Mediterranean diet adherence and its effect on asthma and allergies in children.

Kindly find below my comments for your response.

Abstract

Line 12: Kindly remove “The” from the sentence. Revise to “Dietary pattern may potentially impact on the pathogenesis of asthma and allergies”.

Line 16: replace “in” with “on”. The authors should put a “full stop” after the 26th January, 2022. They should then start a new sentence that indicate “the total number of articles obtained after the initial search on the databases was conducted and state how many were selected after the removal of duplicates and screening for eligibility”

Line 18: The “authors state “…..but no effect was found on allergies”. However, they failed to indicate the magnitude of effect. This systematic review had no meta-analysis component. The authors should indicate how the effect was determined.

Line 18-19: The conclusion can be strengthened. The authors can add the last part of the concluding statement at the “Conclusion section” which is “……highlight the need for randomized controlled trials that will focus on the pediatric population and hopefully provide more robust evidence.” Researchers can then build on this gap that has been identified.

Introduction

Line 26: remove “The”

Line 33: remove “refers” and replace with “…is composed of………..”

Line 39: replace “been focused” with “…..established an association between…..”

Line 46: Did those studies report that outcome in children or in adults?

Materials and Methods

Line 59: The name initials of the two independent reviewers should be indicated. Also, replace “solved” with “resolved”.

Line 63: The authors must revise it to mean: i) Studies that were published in English Language. The authors should add “studies” to the other inclusion criteria listed. Also, which age range did the authors select for inclusion for “children”.

Line 67: The authors should revise the criteria listed to indicate that it was “studies” that investigated those outcomes listed there

Line 72: The name initials of the two authors should be indicated.

Results

Line 77: replace “database” with “databases”.

Line 121: add “size” to the sample

Line 140: The authors must expand every newly introduced abbreviation such as FEV1 and FVC

Discussion

Reference 16, Garcia-Marcos et al. 2013 also conducted a systematic review and meta-analysis. The authors could point this out in the “Introduction” and indicate that this is more of an updated review as this is recent.

Author Response

I uploaded the response to reviewer's comments as a Word file.

Reviewer 2 Report

This is an interesting review addressing the effectiveness of adherence to Mediterranean Diet (MD) against asthma and allergies in childhood. Twelve studies were selected. The authors found a protective role of MD against childhood asthma, but no effect was found on allergies. However, the heterogeneity and limitations of the studies highlight the need for further research.

The review is of interest and of clinical impact. I have only a comment. Regarding the impact of diet on allergies, the authors should recall also the impact of another not rare disease that may link diet and atopy. It has been demonstrated that the prevalence of coeliac disease in atopics is significantly higher than that in the general population, as previously reported (Prevalence of silent coeliac disease in atopics. Dig Liver Dis. 2000 Dec;32(9):775-9. ). Therefore, atopy should be considered a condition at risk of celiac disease requiring gluten-free diet.

Author Response

I uploaded the response to the reviewer's comments as a Word file.
